# Insights on the Larvicidal Mechanism of Action of Fractions and Compounds from Aerial Parts of *Helicteres velutina* K. Schum against *Aedes aegypti* L.

**DOI:** 10.3390/molecules25133015

**Published:** 2020-07-01

**Authors:** Diégina A. Fernandes, Louise H. G. Oliveira, Hyago L. Rique, Maria de Fátima Vanderlei de Souza, Fabíola da Cruz Nunes

**Affiliations:** 1Post Graduation Program in Bioactive Natural and Synthetic Products, Federal University of Paraíba, João Pessoa 58051-900, PB, Brazil; diegina@ltf.ufpb.br (D.A.F.); mfvanderlei@ltf.ufpb.br (M.d.F.V.d.S.); 2Biotechnology Center, Federal University of Paraíba, João Pessoa 58051-900, PB, Brazil; louisehgoliveira@gmail.com (L.H.G.O.); hyagolrique@gmail.com (H.L.R.)

**Keywords:** 7,4′-di-*O*-methyl-8-*O*-sulphate flavone, tiliroside, nitric oxide, cytotoxicity, larvicidal activity

## Abstract

Viral diseases transmitted by the female *Aedes aegypti* L. are considered a major public health problem. The aerial parts of *Helicteres velutina* K. Schum (Sterculiaceae) have demonstrated potential insecticidal and larvicidal activity against this vector. The objective of this research was to investigate the mechanisms of action involved in the larvicidal activity of this species. The cytotoxicity activity of *H. velutina* fractions and compounds of crude ethanolic extract of the aerial parts of this species was assessed by using fluorescence microscopy and propidium iodide staining. In addition, the production of nitric oxide (NO) and hemocyte recruitment were checked after different periods of exposure. The fluorescence microscopy revealed an increasing in larvae cell necrosis for the dichloromethane fraction, 7,4′-di-*O*-methyl-8-*O*-sulphate flavone and hexane fraction (15.4, 11.0, and 7.0%, respectively). The tiliroside did not show necrotic cells, which showed the same result as that seen in the negative control. The NO concentration in hemolymph after 24 h exposure was significantly greater for the dichloromethane fraction and the 7,4′-di-*O*-methyl-8-*O*-sulphate flavone (123.8 and 56.2 µM, respectively) when compared to the hexane fraction and tiliroside (10.8 and 8.3 µM, respectively). The presence of plasmocytes only in the dichloromethane fraction and 7,4′-di-*O*-methyl-8-*O*-sulphate flavone treatments suggest that these would be the hemocytes responsible for the highest NO production, acting as a defense agent. Our results showed that the larvicidal activity developed by *H. velutina* compounds is related to its hemocyte necrotizing activity and alteration in NO production.

## 1. Introduction

*Aedes aegypti* L. (Diptera: Culicidae) is a major vector for viruses that threaten human health, such as dengue, chikungunya, and Zika. Globally, 2.5 billion people live in high-risk areas, especially in tropical and subtropical regions of the world where temperature and humidity promote their proliferation [1,2,3].

Efficient vaccines against these arboviruses have not yet been developed, making vector control the main form of preventing these diseases. Several natural products are considered promising for use as insecticides, repellents, or larvicides, due their biodegradability, efficiency, and low cost [4,5].

According to Faraldo et al. (2005), the larvae possess an extremely efficient immune system that is an excellent model for studying insect defense mechanisms [6]. In contrast to the complexity of the vertebrate immune system, the relative simplicity of the invertebrate immune system makes it a potentially sensitive and accessible means of monitoring the effects of environmental contaminants and the complex interactions that ultimately affect host resistance [7].

Invertebrate hemocytes have been used as a model to study and measure the impact of chemicals on the immune system, including pesticides and heavy materials [8]. Studies in invertebrates have related nitric oxide (NO) production to cytotoxic effects against pathogens, with the increase of its production in hemocytes being correlated with the immune response against foreign agents [9,10].

In this context, *Helicteres velutina* K. Schum (Sterculiaceae) species, popularly known as “pitó” and traditionally used as insect repellent, can be highlighted [11]. Scientific studies have shown its larvicidal activity against *A. aegypti* [11,12,13]. Regarding the phytochemicals from the genus *Helicteres,* previous studies have reported the occurrence of steroids, saponins, and phenolic compounds, such as flavonoids and tannins [13,14].

In order to continue the ongoing work with *H. velutina* against *A. aegypti* larvae, this study investigates the possible mechanisms of action involved in the larvicidal activity of fractions and isolated substances [12,13] by analyzing in vitro cytotoxicity and NO production.

## 2. Results

### 2.1. Larvae (L4) Survival Time after Exposure to Test Substances

The mortality profile of the larvae over time was monitored in order to determine the exposure time to initiate the larvicidal effect.

In this study, using the needed concentration to kill all exposed larvae, it was registered that the dichloromethane fraction of crude ethanolic extract of aerial parts of *H. velutina* (10.0 mg/mL) had the best mortality profile in the first three hours, reaching 33.5%, followed by compound 7,4′-di-*O*-methyl-8-*O*-sulphate flavone (**1**) (1.0 mg/mL), which produced 16.7% mortality, and the hexane fraction (5.0 mg/mL) with 6.7% mortality within three hours. The results for the three tested materials (dichloromethane fraction, hexane fraction, and 7,4′-di-*O*-methyl-8-*O*-sulphate flavone) showed quite similar results over time, reaching 90.0%, 98.34%, and 96.7% after 24 h, respectively (Figure 1).

Tiliroside (**2**) (1.0 mg/mL) showed a later mortality effect, with 6.7% of exposed larvae dead in the first 12 h, reaching 65.0 and 100.0% after 48 and 72 h, respectively. In the negative control, no mortality was recorded (Figure 1).

Comparing the curves, the hexane and dichloromethane fractions of the crude ethanolic extract of aerial parts of *H. velutina* do not differ statistically. The hexane fraction differs from tiliroside and 7,4′-di-*O*-methyl-8-*O*-sulphate flavone, and the dichloromethane fraction differs from tiliroside, but does not differ from 7,4′-di-*O*-methyl-8-*O*-sulphate flavone. Tiliroside and 7,4′-di-*O*-methyl-8-*O*-sulphate flavone differ from each other (Table 1).

Considering the used doses, the most active substance would be 7,4′-di-*O*-methyl-8-*O*-sulphate flavone, which showed the greatest mortality profile at 1.0 mg/mL.

Morphological changes and macroscopic aspects showed that the larvae treated with the hexane fraction were more debilitated, allowing us to conclude that although the larvicidal action is not as expressive in the first hours as in the dichloromethane fraction and 7,4′-di-*O*-methyl-8-*O*-sulphate flavone, it is more aggressive after 24 h of exposure. The greater effectiveness of the hexane fraction can also be seen since it is necessary to double the dichloromethane fraction concentration to trigger larvicidal activity in the same time interval. Tiliroside did not show visible body changes, but a slight change in the larvae body color was observed (Figure 2).

### 2.2. Measurement of Nitric Oxide

The concentration of NO in the larvae treated with the hexane fraction was higher after 3 h of exposure (37.5 ± 6.5 µM) and decreased at 6 and 24 h (27.3 ± 7.5 and 10.8 ± 3.0 µM), representing 100%, 72.8%, and 28.8% NO in this sample (Figure 3A-1).

The dichloromethane fraction showed the opposite result. The concentration of NO was 29.2 ± 0.6 µM initially, increasing significantly over time to 80.1 ± 12.7 and 123.8 ± 27.1 µM in 6 and 24 h, corresponding to 23.6, 64.7, and 100%, respectively (Figure 3A-2).

The compound 7,4′-di-*O*-methyl-8-*O*-sulphate flavone (**1**) showed similar results to the dichloromethane phase, although NO production was less significant. The NO concentration was 21.9 ± 1.8 and 26.2 ± 3.0 µM after 3 and 6 h of exposure, respectively, reaching 56.2 ± 8.8 µM in 24 h, with a percentage of 38.9, 46.6, and 100% (Figure 3B-1).

The tiliroside (**2**), as well as the hexane fraction, had a maximum peak of 78.9 ± 8.0 µM NO after 3 h of exposure, decreasing to 23.4 ± 5.2, 8.3 ± 1.2 µM and remaining at 20.5 ± 4.6 µM in the last hours of the assay (100%, 29.6%, 10.5%, and 26.0% (Figure 3B-2).

The substances that have the best action over time are 7,4′-di-*O*-methyl-8-*O*-sulphate flavone and the dichloromethane fraction. Although sulphated flavonoid has shown a tendency to kill a higher percentage over time, statistically it has an action similar to its original fraction (dichloromethane). Among the tested substances, the one that had the least effect compared to the dose used was tiliroside, taking longer to kill the larvae.

### 2.3. Cytotoxic Activity

After analyzing the images obtained by fluorescence microscopy, it was possible to observe that within 24 hours of exposure, there was no significant cellular necrosis of the hemocytes of the larvae treated with the tiliroside when compared with the hemocytes of the control group. Greater propidium iodide (PI) impregnation was observed for cells treated with the dichloromethane fraction, 7,4′-di-*O*-methyl-8-*O*-sulphate flavone, and hexane fraction, with a percentage of hemocyte necrosis of 15.4%, 11.0% and 7.0%, respectively (Figure 4).

The cell types found in each treatment were identified in order to correlate the results obtained with the possible mechanisms of action involved in cell death. Morphological analyses included observation of region geometries and obtaining data of interest [15]. The identification of hemocytes was performed based on comparisons with the literature [16], and the diameter and area were measured using the ImageJ software.

Table 2 lists the number of total and viable cells counted in the hemocytometer with computational images and ImageJ software. The main uses of the program include optimization, algebraic manipulation, counting, and defining the areas and diameters of the cells under analysis.

With the data obtained in Table 2, it was also possible to calculate the number of necrotic cells in each experiment (Figure 4). The dichloromethane fraction presented a higher percentage of necrotic cells (stained) in relation to the total number of cells counted (15.4%). The 7,4′-di-*O*-methyl-8-*O*-sulphate flavone (sulphated flavonoid) had a percentage necrosis of 11.0%, followed by the hexane fraction (7.0%) and the tiliroside (glucoside flavonoid) did not show necrotic cells, which showed very similar result to that seen in the control. There was no statistical difference between the groups.

In the present study, the hemocytes found in the analyses were oenocytoids, prohemocytes, and plasmatocytes. The prevalence of each hemocyte type for the test substances are shown in Figure 5.

The oenocytoids were predominant in Tiliroside and negative control (51.6 and 100.0%, respectively), prohemocytes in turn were majority in the hexane and dichloromethane fractions, and in the substance 7,4′-di-*O*-methyl-8-*O*-sulphate flavone (96.0%, 59.9% and 73.8%, respectively).

The plasmatocytes were only found in the dichloromethane fraction and 7,4′-di-*O*-methyl-8-*O*-sulphate flavone (7.4 and 13.1%, respectively), the presence of this hemocyte suggests that the larvae need to recruit more defense agents to combat these substances, which may characterize their greater toxicity in the larvae, corroborating what was seen previously in the survival tests and NO production in the larvae of *A. aegypti* exposed to these substances.

## 3. Discussion

In Brazil and in other parts of the world, plant extracts (medicinal, native, and adapted) and their derivatives have been tested against different disease vectors, including *A. aegypti* and *A. albopictus* [17]. The species *H. velutina* stands out for its popular use as repellent [11] and for its proven larvicidal activity of the extract, fractions, and isolated substances against *A. aegypti* [12,13].

Monitoring larval mortality over time has provided data to better understand the possible mechanisms of action involved in the larvicidal activity. It has been shown that the dichloromethane fraction, hexane fraction, and the substance 7,4′-di-*O*-methyl-8-*O*-sulphate flavone trigger a faster larvicidal activity compared to tiliroside.

Studies on this survival profile are still scarce; however, Nunes (2013) reported that *Agave sisalana* had a similar larvicidal action over time using the concentration of 6.5 mg/mL. For *A. sisalana*, the larvae started dying after 12 h, reaching 88.0% larval mortality at 24 h of exposure [18].

During this assay, the macroscopic aspects of the larvae and their morphological changes were also observed after 24 h, allowing us to verify that in the test groups, even when the larvae were not dead, they were weakened, with reduced motility, and showing alteration in color and body aspects. In the negative control group, the larvae had normal external morphology and motility [19].

Larvae treated with the hexane fraction showed greater body deterioration. On the other hand, the dichloromethane induced a contraction of larvae body. The substances induced toxic effects on many regions of the body (including thorax, abdomen, anal gills, loss of external hairs, crumbled epithelial layer of the outer cuticle, and shrinkage of the larvae), results similar to those found in other studies [20,21].

These data also help explain the greater effectiveness of the hexane fraction, which requires half the concentration of the dichloromethane fraction to have the same percentage of mortality within 24 hours of exposure. The larvae exposed to the sulphated flavonoid (7,4′-di-*O*-methyl-8-*O*-sulphate flavone) showed greater deterioration and loss of color, being more opaque than those exposed to tiliroside, suggesting that these substances act with different mechanisms of action.

NO production was quantified as nitrite ion in the hemolymph of *A. aegypti* larvae (L4). For tiliroside, the NO was evaluated after 72 h because it developed later mortality. For the other test substances, the 24 h interval was used [13].

The hexane fraction and tiliroside showed a similar NO production profile, where after a maximum peak in the first hours of exposure, the concentration decreased over time. This decrease may be related to the slower action of these compounds on the larvae. This later response could cause a smaller number of hemocytes to be recruited and low production of defense cells, which would result in a lower concentration of NO and a greater action of the aggressor agent.

Meanwhile, the dichloromethane fraction and the 7,4′-di-*O*-methyl-8-*O*-sulphate flavone developed the opposite result on NO concentration, increasing over time. The result on NO concentration was more significant for the dichloromethane fraction, reaching 123.8 µM. This higher NO production may be related to a greater recruitment of hemocytes to act against the larvicidal agent, since these substances start to trigger mortality in the first hours, being more aggressive. Our results show that all substances tested cause a significant increase in the NO levels of larvae. These levels are many times higher than the basic levels of the controls (64.5, 213.4, 96.4, and 136 times for hexane, dichloromethane, sulphate, and tiliroside, respectively, in 24 h). The excess of NO is potentially toxic, especially with regard to oxidative stress, as demonstrated by Oliveira et al. (2016) [22].

Nunes et al. (2015) have reported that the increase in NO production by larvae hemocytes is correlated with the immune response against foreign agents [9]. Previous studies have reported an increase in NO production by invertebrates over time, showing the involvement of this molecule in the insect’s immune defense [6,10,23].

The determination of the total cell number and their viability are considered important measures to study the mechanisms of action of substances in insects and larvae [18]. There are several ways to measure cell viability, the most common being the detection of membrane integrity. Defective membranes allow the releasing of intracellular components that may be found in extracellular fluids [15].

In our study, the propidium iodide (PI) was used as a marker of cell necrosis, which crosses only necrotic cell membranes, staining DNA and RNA present in the cytoplasm. PI emits red fluorescence when absorbing UV light [18,24,25].

The results showed greater PI impregnation by cells treated with the dichloromethane fraction, 7,4′-di-*O*-methyl-8-*O*-sulphate flavone, and hexane fraction, respectively. No necrosis was detected in the cells of the larvae treated with tiliroside; slower activity of this substance can cause changes in cell responses. These data corroborate the fact that the dichloromethane and hexane fractions and the sulphated flavonoid recruit a greater number of defense cells, while the tiliroside attract insufficient number of cells, justifying the lower NO production.

There are few studies that evaluate the percentage of cell necrosis through fluorescence microscopy and staining with PI for hemocytes of *A. aegypti* larvae. In a previous investigation, Nunes et al. (2015) used flow cytometry to analyze the percentage of hemocyte necrosis within 3, 6, 12, and 24 h of exposure to *A. sisalana*. It has showed that in the first 12 h of exposure of the larvae there was 21% of necrosis, and in 16 h of 16.5% compared to the control group [9].

The free circulating cells in the hemolymph are called hemocytes and have different forms and functions [18]. The number and types of hemocytes can vary in response to stress, injuries, and infections. After an injury or contact with toxic compounds, the hemocytes migrate to the place where they destroy the invading agents, as part of the insect’s defense mechanisms, including recognition, phagocytosis, encapsulation, coagulation, nodule formation, and cytotoxicity [16].

In the present study, the hemocytes types found were: oenocytoids, prohemocytes, and plasmatocytes. Oenocytoids measure 7–10 µm in diameter, possess a round shape, and have small, lobulated, and eccentric nuclei. Prohemocytes are the smallest cells found in hemolymph. They usually are found in groups, with spherical, oval, or even elongated shapes, measuring 5–7 µm in diameter. They are very similar to oenocytoids, when observed in the Neubauer chamber, differing by size. Plamatocytes are very polymorphic cells, ranging from rounded to elongated with 9–40 µm in diameter [16].

According to Araújo (2011), for the mosquito *A. aegypti*, the most prevalent hemocyte subpopulations are prohemocytes (53.3%), followed by oenocytoids (30.8%) and plasmatocytes (12%). They form up to 96% of the total cells. Trobocytoids, adipohemocytes, and granulocytes are the other hemocyte kinds [16].

In our study, it was possible to observe that the plasmocytes were only observed in the larvae exposed to the dichloromethane fraction and 7,4′-di-*O*-methyl-8-*O*-sulphate flavone, the two treatments that presented the highest percentage of necrosis. Previous studies have shown that granulocytes and plasmatocytes are the main hemocytes actively involved in cellular defenses of insects [26]. Greater production of prohemocytes was observed for 7,4′-di-*O*-methyl-8-*O*-sulphate flavone and the dichloromethane and hexane fractions, and the oenocytoids were predominant in larvae treated with tiliroside. 

The cell viability data led us to suggest that the NO produced by the larvae treated with the hexane fraction and tiliroside would not be enough to generate an effective response against the toxic substances and would decay over time, resulting in the larvae mortality. The plasmatocytes found when the larvae were treated with the dichloromethane fraction and the 7,4′-di-*O*-methyl-8-*O*-sulphate flavone suggest that the compounds were recognized as a more aggressive toxic agents. Thus, other defense cells were recruited and start to produce excessive NO to generate an effective response against the toxic larvicidal substances. The greater the number of cells killed by necrosis, the more NO the remaining cells will produce as a defense mechanism [6].

Both flavonoids tested here were isolated from the dichloromethane fraction [12], according to the results obtained. The 7,4′-di-*O*-methyl-8-*O*-sulphate flavone has the most promising activity of the fraction, different from tiliroside, which triggered later activity and resembled the results found in the negative control. Our findings corroborate those that were previously reported about the importance of the sulphate group (OSO_3_H) for larvicidal activity [13].

## 4. Materials and Methods

### 4.1. Plant Material

The aerial parts of *H. velutina* were collected in Serra Branca/Bahia/Brazil, in the winter season. The material was identified by Prof. Adilva de Souza Conceição, and a specimen voucher was kept in the Herbarium of the State University of Bahia. The plant material was oven dried at 40 °C, and 1976.0 g of the powder was macerated with 95% ethanol for 72 h. The extract solution was dried under reduced pressure at 40 °C and provided 39.7 g of crude ethanolic extract (CEE) that was submitted to liquid–liquid chromatography using hexane, dichloromethane, ethyl acetate, and *n*-butanol, resulting in their respective fractions and a hydroalcoholic fraction [12].

The biomonitored, phytochemical study of extracts derived from the aerial components of *H. velutina* against *A. aegypti* larvae resulted in the discovery of the promising biological activities of the hexane and dichloromethane fractions. From those fractions, chromatographic and spectroscopic methods resulted in the isolation and identification of 17 substances [12,13]. Two isolated flavonoids from dichloromethane fraction, tiliroside (glucoside flavonoid) and 7,4′-di-*O*-methyl-8-*O*-sulphate flavone (sulphated flavonoid) (Figure 6), were submitted to in vitro assays, which demonstrated that these compounds showed larvicidal activities at low concentrations [13].

This study has been registered in the National System of Genetic Resource Management and Associated Traditional Knowledge (SisGen—A568B8A).

### 4.2. Larvae Survival Time Exposed to Test Substances

The fourth-stage *A. aegypti* larvae (L4) (Rockefeller strain) were obtained from the Laboratory of Biotechnology Applied to Parasites and Vectors, Biotechnology Center, Federal University of Paraiba. They were kept under conditions of biological oxygen demand (BOD), at 27 ± 2 °C, relative humidity of 27 ± 5 °C, with a photoperiod of 12 h light and 12 h dark [13].

The survival time of the larvae exposed to the test substances was evaluated at intervals of 0, 3, 6, 12, 24, 48, and 72 h. The concentrations used for the hexane fraction (5.0 mg/mL), dichloromethane fraction (10.0 mg/mL), and for both isolated compounds, 7,4′-di-*O*-methyl-8-*O*-sulphate flavone and tiliroside (1.0 mg/mL), were determined experimentally [13]. To evaluate the insecticidal activity of the different substances over the 72 h period, a comparison analysis of the survival curve was performed using the Log Rank (Mantel-cox) and Chi square test using the Prism program.

### 4.3. Measurement of Nitric Oxide

The production of nitric oxide (NO) was determined by the Griess reagent in a pool of hemolymph from 20 larvae (L4) in a final volume of 20 µL in PBS buffer [27]. The larvae were exposed to the LC_50_ of the hexane fraction, dichloromethane fraction, and 7,4′-di-*O*-methyl-8-*O*-sulphate flavone and tiliroside for different periods of time (3, 6, and 24 h). The control groups were only exposed to distilled tap water and 1% DMSO [28] for the same periods of time. The assays were performed in triplicate. To determine NO_2_^−^ concentrations, an aliquot of each sample was analyzed using spectrophotometry. The absorbance was measured using a microplate reader with a 562 nm filter, and the NO was quantified using a standard curve of NaNO_2_ as reference.

Statistical analyses were performed using GraphPad Prism software for Windows version 5.0 (GraphPad Software, San Diego, CA, USA). Significant differences among groups were analyzed by analysis of variance (ANOVA) followed by the Tukey’s post hoc test when appropriate (*P* < 0.05).

### 4.4. Cytotoxicity Assay

Fluorescence microscopy was performed with a pool of hemolymph of 20 larvae (L4) exposed to LC_50_ of the fractions and substances isolated from *H. velutina* for 24 h.

The larvae were washed in PBS buffer and immobilized under refrigeration (1–2 min). Then, they were placed in a petri dish and, with the aid of a magnifying glass and a scalpel blade, they were decapitated. The hemolymph was collected using a glass microcapillary and transferred into a 1.5 mL Eppendorf containing 100 µL of PBS buffer.

The hemolymph pool was then centrifuged under refrigeration (4 °C) at 1500 rpm for 10 min. The supernatant was discarded and 20 µL of the cell button was transferred to another Eppendorf containing 160 µL of PBS. Then, 20 µL of Propidium Iodide (PI) was added to differentiate intact hemocyte from that in necrosis. The sample was then incubated for 15 min in the dark. Using a micropipette, a 10 µL aliquot of the sample was placed in the Neubauer chamber, the cell integrity and viability were analyzed using a fluorescence microscope using the 20× objective [15].

The cells were characterized using morphology and size as parameters. Prohemocytes are spherical, oval, or even elongated, measuring about 5–7 µm in diameter. Adipohemocytes are round or oval cells that measure approximately 12–50 µm in diameter. They have a round nucleus that is centralized or displaced to the periphery of the cell. Its cytoplasm is quite characteristic with the presence of large lipid vesicles. Granulocytes have a circular shape, with 8–20 µm diameter. They have an irregular plasma membrane and the cytoplasm exhibits some dense granules. Plasmatocytes are very polymorphic cells, 9–40 µm in diameter. The plasma membrane has an irregular surface showing philopodia and pseudopods, with characteristics of fibroblasts. Oenocytoids measure 7–10 µm in diameter, have a round shape, with small, lobulated, and eccentric nucleus. The ultrastructure reveals a nucleus without the presence of a prominent nucleolus and homogeneous cytoplasm with few organelles. The ImageJ software was used to measure the diameter and area of the cells in order to characterizing them [16].

The total concentration of cells is given by the sum of the number of viable cells (not stained) plus the number of nonviable cells (stained) and multiplied by the dilution factor. Equations (1)–(3) show the calculations of the total cell concentration, as well as the concentration and percentage of viable cells. Equation (4) was used to calculate the number of necrotic cells.

**Equation (1).** Total cell concentration:(1)$$\left(n_{V}+n_{D}\right)\times \;D\;\times 10^{4}=\mathrm{cells}/\mathrm{mL}$$

**Equation (2).** Concentration of viable cells:(2)$$n_{V}\times \;D\;\times 10^{4}=\mathrm{cells}/\mathrm{mL}$$

**Equation (3).** Percentage of viable cells: (3)$$\frac{n_{V}}{n_{V}+n_{D}}\times 100=\%\;\mathrm{viable}\;\mathrm{cells}$$

**Equation (4).** Number of necrotic cells:(4)$$\frac{n_{V}+n_{D}}{n_{C}}\times \;D\;\times 10^{4}=\mathrm{cells}/\mathrm{mL}$$
where $n_{V}$ corresponds to the total number of viable cells, $n_{D}$ is the total number of nonviable cells, D is the dilution factor (in our case D = 10), and $n_{C}$ is the number of quadrants counted on the Neubauer chamber (in our case $n_{C}=4$).

## 5. Conclusions

The present study contributed to the knowledge of the biological action of the species *H. velutina* and its compounds against *A. aegypti* larvae. The mechanisms of action responsible for causing the death of mosquito larvae are complex and generally multifactorial. Our results show that all substances tested cause a significant increase in the NO levels of larvae. With regard to cytotoxicity, we found that only tiliroside does not cause cell necrosis. Thus, we can conclude that an increase in NO levels plays a key role in the mechanism of action of the larvicidal activity of *H. velutina*. This effect is potentiated by the necrotizing action of all substances, except tiliroside.

## Figures and Tables

**Figure 1 molecules-25-03015-f001:**
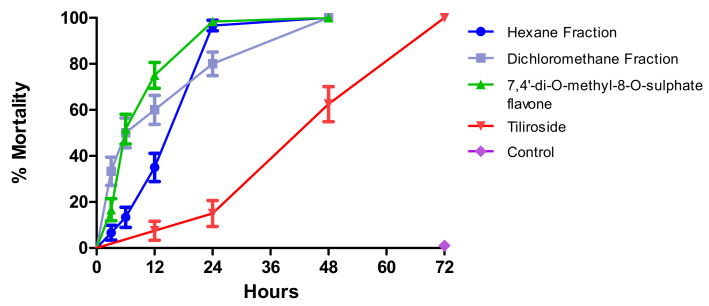
Percentage of larval survival over time of exposure to test substances.

**Figure 2 molecules-25-03015-f002:**
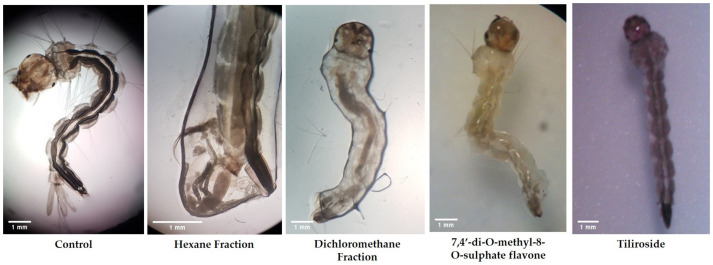
Stereomicroscopic view of fourth instar of *A. aegypti* larvae after 24 h of exposure. Substances induced toxic effects on many regions of the body (including thorax, abdomen, anal gills, loss of external hairs, crumbled epithelial layer of the outer cuticle, and shrinkage of the larvae).

**Figure 3 molecules-25-03015-f003:**
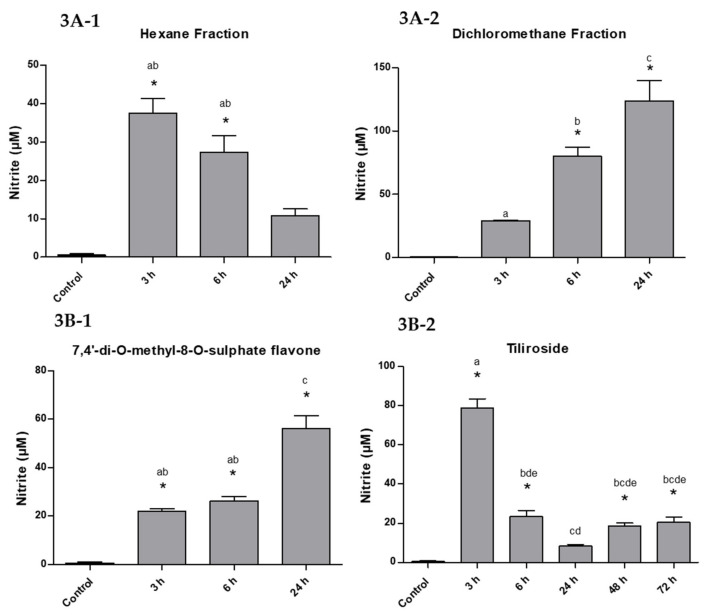
NO production in larvae exposed to LC_50_ of the substances over time. (*) statistically significant in relation to the negative control. Bars with the same letter are not significantly different by Tukey test, 5%.

**Figure 4 molecules-25-03015-f004:**
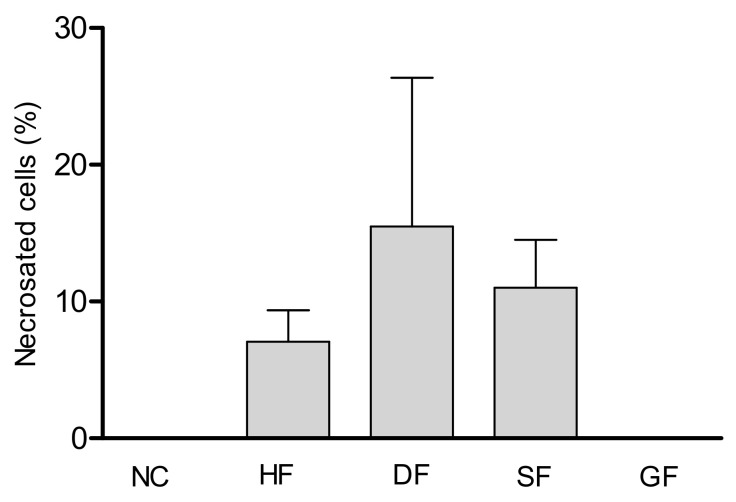
Percentage of cell necrosis observed. HF: Hexane Fraction; DF: Dichloromethane fraction; SF: Sulphated Flavonoid, GF: Glucoside Flavonoid, and NC: Negative Control.

**Figure 5 molecules-25-03015-f005:**
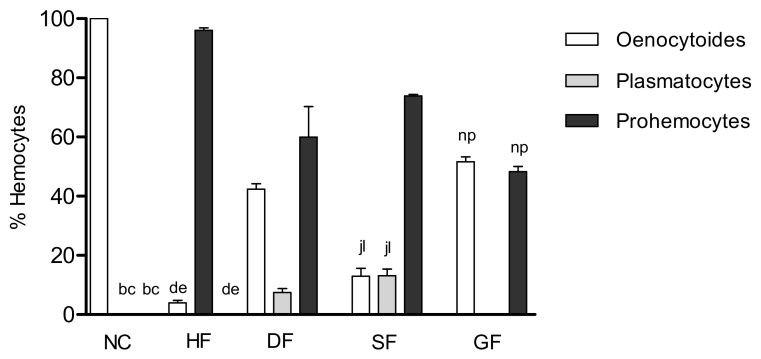
Hemocytes present in larvae exposed to substances. HF: Hexane Fraction; DF: Dichloromethane fraction; SF: Sulphated Flavonoid, GF: Glucoside Flavonoid and NC: Negative Control. Bars with the same letter are not significantly different by Tukey test, 5%.

**Figure 6 molecules-25-03015-f006:**
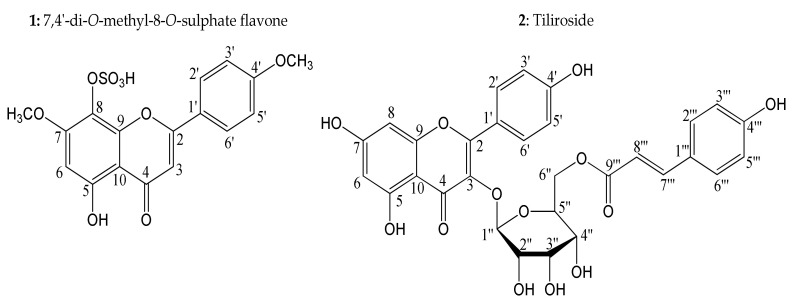
Compounds isolated from aerial parts of *H. velutina* with activity against larvae (L4) of *A. aegypti*.

**Table 1 molecules-25-03015-t001:** Statistical data obtained from survival analysis.

Comparison of Survival Curves						
	HF and DH	HF and SF	HF and GF	DF and SF	DF and GF	SF and GF
**Chi square**	1.498	21.13	57.72	2.977	49.51	73.52
**df**	1	1	1	1	1	1
**P value**	0.2210	<0.0001	<0.0001	0.0845	<0.0001	<0.0001

HF: Hexane Fraction; DF: Dichloromethane fraction; SF: Sulphated Flavonoid, GF: Glucoside Flavonoid. df: degree of freedom.

**Table 2 molecules-25-03015-t002:** Cells counted in the Neubauer chamber, using a fluorescence microscope.

Test Substances	Total of Cells/mL	N° of Viable Cells/mL	% of Viable Cells
Control (−)	1.2 × 10^6^	1.2 × 10^6^	100.0%
Hexane fraction	1.32 × 10^6^	1.3 × 10^6^	98.1%
Dichloromethane fraction	5.5 × 10^6^	5.15 × 10^6^	93.6%
Tiliroside	9.2 × 10^5^	9.2 × 10^5^	100.0%
7,4′-di-*O*-methyl-8-*O*-sulphate flavone	2.5 × 10^6^	2.46 × 10^6^	98.7%

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
