# Peer review of "Insights on the Larvicidal Mechanism of Action of Fractions and Compounds from Aerial Parts of Helicteres velutina K. Schum against Aedes aegypti L."

_molecules, 2020, doi:10.3390/molecules25133015_

Round 1

Reviewer 1 Report

Reviewer: 1

Ms. Ref. No.: molecules-832447

Authors: Diégina A. Fernandes, Louise H. G. Oliveira, Hyago L. Rique, Maria de Fátima Vanderlei de Souza, Fabiola da Cruz Nunes

Specific notes:

TITLE

Nothing to comment.

ABSTRACT

Line 15: Delete “we” and rewrite the sentence

Line 28: Helicteres velutina must be abbreviated

KEYWORDS

Line 30: The keywords do not seem to be keywords, since two of them also appear in the title, please change/modify it ...

INTRODUCTION

Line 41: Delete “we” and rewrite the sentence

Line 42: ¿Ae. Aegypti?, please correct

Line 46: ¿Ae. Aegypti?, please correct

RESULTS

The authors speak in the results section of significant differences, but do not show at any time the statistical data obtained in the analyzes (F, df, P ..). Please put great emphasis on adding the suggested data to explain this section.

Line 140: ¿Ae. Aegypti?, please correct

DISCUSSIÓN

Lines 147 and 148: ¿Ae. Aegypti and Ae. albopictus?, please correct

Line 149: ¿Ae. Aegypti?, please correct

Line 172: ¿Ae. Aegypti?, please correct

Line 173: Delete “we” and rewrite the sentence

Lines 193 and 194: ¿Ae. Aegypti?, please correct

Line 202: Delete “we” and rewrite the sentence

Line 212: ¿Ae. Aegypti?, please correct

Line 228: ¿Ae. Aegypti?, please correct

Line 249: Delete “we” and rewrite the sentence

MATERIALS AND METHODS

(4.1. Plant material), from the line 255 to the line 262, the authors should provide more information on the extraction of the compounds from H.velutina and the obtaining of the L4 larval stages of A. aegypti

Line 260: ¿Ae. Aegypti?, please correct

(4.1. Plant material), from the line 317 to the line 331, the letters of the equations are blurred, please clarify them.

FIGURES

Nothing to comment.

TABLES

Nothing to comment.

Author Contributions

Line 345: Complete with the contributions of each author

REFERENCES

Review, as there are fully underlined citations, just like DOIs. Please consult the author's guidelines of the journal, and unify

Author Response

Dear reviewer,

We truly appreciate your comments and incorporate your suggestions into the manuscript. Please see below for our answers to each of your recommendations or questions. Thank you very much for your contribution in improving the manuscript.

Best wishes,

Reviewer 2 Report

The manuscript Molecules ID: molecules-832447 - describes and proposes the use of the aerial parts of Helicteres velutina K. Schum (Sterculiaceae) as insecticidal and larvicidal activity against the female Aedes aegypti L. considered a vector for viral disease.

COMMENT

Lane 56 incorrectly reports Figure 5, but this is the first figure in the text and cited before Figure 1 in lane 60

Lane 275 cited the Figure 5 that refers the isolated compounds structure

Lanes 17-107- 202 report the cellular necrosis/cytotoxic activity investigated by using fluorescence microscopy and propidium iodide staining, but the propidium iodide (PI) impregnation method used by the Authors is not described in the method section. The paragraph 202-215 in the discussion section is not sufficient.

The research describes the larvicidal activity of the Helicteres velutina, but the manuscript never reports or describes how these larvae are obtained, grown and selected for the experiments. Since the larvae are the focus of the research, the authors must add these in the methods section.

Lane 112- Authors are invited to describe in the Methods section, as they performed the identification of hemocytes (references are not sufficient) and measured the diameter and area of the hemocytes.

Moreover at lane 157, Authors report “.. the macroscopic aspects of the larvae and their morphological changes”, but no images are shown to evaluate the morphology.

In the manuscript there are some references without DOI, and not readable, which for this reason they cannot provide the necessary information for understanding and evaluating the research.

Lane 193 - Authors discussed about a possible inhibitory effect of the enzyme NO synthase. To shed light on this question they can performed analysis on the expression of NO synthase enzyme by RT-PCR or Western-blotting

4.1 Plant material - This section needs to be implemented by adding which aerial part is used, the season and the site of collection. The extraction method requires additional information in addition to the references.

The section 4 should be improved in order to better accompany the data reported in section 2. Results and 3.Discussion

The manuscript needs major revision.

Author Response

(The authors gave the same response as above.)

Reviewer 3 Report

Major comments: 1. Article is very difficult to follow for readers. Authors should improve introduction, result and discussion section. 2. Result section does not include extraction and isolation of compounds from Helicteres velutina K. Schum. Moreover Authors did not characterise the isolated compounds. 3. Page 4, line 104 and 105: After analysing the images obtained by fluorescence microscopy, it was possible to observe that, within 24 h of exposure, there was no significant cellular necrosis of the hemocytes of the larvae treated with the tiliroside when compared with the hemocytes of the control group: Authors have not include any representative image to support their results. 4. Authors concluded that that there are different mechanisms of action for these larvicidal substances from H.velutina, but the results do not fully support their conclusion. Require more experiments to show different mechanisms.

Author Response

(The authors gave the same response as above.)

Round 2

Reviewer 1 Report

Dear Dr. Calin Ovidiu Cadar
Assistant Editor, MDPI
MDPI Open Access Publishing Romania
Str Avram Iancu 454, 407280 Floresti, Cluj, Romania
Molecules Editorial Office
Skype: calin.cadar90
E-mail: [email protected]; [email protected]

Manuscript ID molecules-832447-v2 entitled "Insights on the larvicidal mechanism of action of fractions and compounds from aerial parts of Helicteres velutina K. Schum against Aedes aegypti L.

Reviewer: 1

Ms. Ref. No.: molecules-832447-v2

Authors: Diégina A. Fernandes, Louise H. G. Oliveira, Hyago L. Rique, Maria de Fátima Vanderlei de Souza, Fabiola da Cruz Nunes

General note:

Thank you very much to the authors for having added all the suggested comments.

Reviewer 2 Report

The manuscript Molecules ID: molecules-832447 - describes and proposes the use of the aerial parts of Helicteres velutina K. Schum (Sterculiaceae) as insecticidal and larvicidal activity against the female Aedes aegypti L. considered a vector for viral disease.

Authors took the suggestions into account and answered all the questions.

There are some errors in the text and some abbreviations are reported differently

The  revised manuscript has been improved by the Authors and it can  be accepted for publication.

Reviewer 3 Report

Authors have responded all comments properly and have improved manuscript to be consider for publication